# CellFlows: Inferring Splicing Kinetics from Latent and Mechanistic Cellular Dynamics

Sei Chang [* 1 2]  Zaiqian Chen [* 1 2]  Bianca Dumitrascu [1 3]  David A. Knowles [1 2]

## Abstract

RNA velocity-based methods estimate cellular dynamics and cell developmental trajectories based on spliced and unspliced RNA counts. Although numerous methods have been proposed, the underlying models vary greatly in their biophysical assumptions, architectures, and use cases. In this work we introduce a new architecture, *CellFlows*, which incorporates self-supervised neural dimensionality reduction with the flexibility of neural-based latent time estimation into a mechanistic model, improving model interpretability and accuracy. *CellFlows* models splicing dynamics to infer gene and context-specific kinetic rates at single-cell resolution and correctly identifies both linear and branching cellular differentiation pathways originating from mouse embryonic stem cells.

## 1. Introduction

The biological processes driving cellular changes in development and disease are inherently dynamic. The most canonical dynamic process is the differentiation of pluripotent stem cells, which can give rise to any of the functionally diverse cells that compose a metazoan. Although single-cell RNA sequencing has enabled the recovery of cellular heterogeneity by capturing gene expression profiles, the protocol only provides a single snapshot of the transcriptome for a given sample, making it difficult to infer the underlying cellular dynamics experimentally.

To address this challenge, computational techniques have been devised to infer the ordering of cells along a developmental trajectory based on their transcriptomic profiles. Prior trajectory inference methods characterize cellular de-

velopment as occurring along a continuous "pseudotime" axis between 0 and 1, based on similarities of gene expression profiles between single cells (Haghverdi et al., 2016). While pseudotime effectively serves as a one-dimensional coordinate to arrange cells along a single lineage, it lacks the expressivity required to capture the full complexity of splicing dynamics and multiple lineages. Recently developed methods have incorporated a system of linear ordinary differential equations (ODEs) (La Manno et al., 2018). The linear ODEs determine the gradient of spliced and unspliced RNA through interpretable kinetic rates that describe transcription, splicing, and decay. By leveraging the causal relationship between spliced and unspliced RNA abundances in single-cell data, methods based on the framework estimate "RNA velocity", the rate of change of spliced counts $ds/dt$. Inferring the RNA velocity vector across all genes would enable the prediction of future RNA abundance.

While the linear ODEs provide a mechanistic basis for RNA velocity to model cellular dynamics, RNA velocity methods apply the framework with varying assumptions inconsistent with known underlying biophysics (Gorin et al., 2022). The first RNA velocity model for single cells, velocyto (La Manno et al., 2018), uses a steady-state model to estimate the RNA velocity from spliced and unspliced counts, assuming constant splicing rates across genes and cells. In contrast, other methods such as scVelo use a fully dynamical model of RNA kinetics (Bergen et al., 2020). Some methods even eschew mechanistic equations in favor of purely neural net-based black box approaches (Li, 2023).

The computational methods that aim to apply the RNA velocity framework to infer and predict developmental trajectories vary significantly in their architectures, parameter assumptions, and mechanistic formulations. Here, we propose *CellFlows*, a novel architecture that infers mechanistic cell and gene-specific transcription, splicing, and decay kinetics to regularize the latent representations learned through a variational autoencoder (VAE) and neural ordinary differential equations (ODEs). We compare the model's performance on publicly available single-cell RNA sequencing (scRNA-seq) datasets that capture the dynamics of mouse embryonic stem cell differentiation and gastrulation erythroid maturation against recently published methods.

[*]Equal contribution  [1]Department of Computer Science, Columbia University, New York City, NY, USA [2]New York Genome Center, New York City, NY, USA [3]Irving Institute for Cancer Dynamics, Columbia University, New York, NY, USA. Correspondence to: Bianca Dumitrascu <bmd2151@columbia.edu>, David A. Knowles <dak2173@columbia.edu>.

*Accepted at the 1st Machine Learning for Life and Material Sciences Workshop at ICML 2024.* Copyright 2024 by the author(s).

## 2. Related Work

**Learning cell-gene splicing kinetics**. cellDancer (Li et al., 2023) uses a "relay" model to infer velocity locally for each cell based on its neighbors, rather than assuming the same kinetics globally for all cells. Local velocity inference based on gene-specific kinetic parameters enables the model to handle heterogeneous cell populations. Prior velocity methods such as scVelo (Bergen et al., 2020) rely on strong biological assumptions such as uniform gene-specific kinetics across all cells, which fails to account for differences in kinetics between cell subpopulations. cellDancer addresses the limitations directly by predicting cell and gene-specific transcription, splicing, and degradation kinetic rates using a multi-layer perception (MLP). The kinetic rates parameterize the RNA velocity through a system of linear ODEs:

$$\begin{aligned}
\frac{du}{dt} &= \alpha(t) - \beta(t)u(t) \\
\frac{ds}{dt} &= \beta(t)u(t) - \gamma(t)s(t)
\end{aligned} \tag{1}$$

where $t$ is time, $u(t)$ is the unspliced abundance, $s(t)$ is the spliced mRNA abundance, $\alpha(t)$ is the transcription rate, $\beta(t)$ is the splicing rate, and $\gamma(t)$ is the decay rate, all at time $t$. The MLP is trained with a loss function based on velocity vector similarity to its $k$-nearest neighbors in gene expression. cellDancer uniquely learns biologically interpretable parameters to constrain its black-box nonlinear approximation of the dynamics.

**Joint inference of latent space and latent dynamics.** scTour (Li, 2023) jointly infers developmental pseudotime, a transcriptomic vector field, and latent representations of cells. Unlike cellDancer and other RNA velocity methods, scTour does not consider splicing dynamics and instead leverages its unique model architecture to derive cell representations. scTour applies a VAE to denoise and extract a latent representation from the single-cell data. The model simultaneously uses a second encoder to extract a pseudotime label for each cell. The latent state of the cell with the earliest pseudotime, along with the encoded pseudotimes of all cells, are fed directly into a neural ODE to predict future gene expression-based cell states. scTour then aligns the neural ODE-generated predictions with the VAE-based latent representations of the measured cells to find the optimal vector field that describes the developmental trajectory.

**Key Limitations**. cellDancer learns its transcriptomic vector field and cell-specific pseudotime in a two-stage process similar to scVelo (Li et al., 2023). In the first stage, cellDancer's MLP is trained to learn RNA velocity and interpretable kinetics. In the second stage, the velocity estimates are used to construct cell-to-cell transition probabilities for inferring pseudotime. This approach does not explicitly integrate pseudotime into the deep learning model. On the other hand, scTour learns the vector field and pseudotime jointly

in training using a neural ODE to overcome these issues, but avoids learning dynamics from spliced and unspliced abundances in favor of total gene expression (Li, 2023).

## 3. CellFlows Architecture

Consider a scRNA-seq dataset $\mathcal{D} = \{X_i\}_{i=1}^n$, where $X_i \in \mathbb{R}^{2g}$ corresponds to the concatenated $g$-dimensional read counts of both spliced and unspliced RNA. $n$ is the total number of captured cells and $g$ is the total number of genes. Our dataset consists of RNA-seq snapshots taken across multiple time points, and we consider $\tau_i \in \mathbb{R}$ as the true process time (in days) for each cell $i$.

### 3.1. Variational autoencoder and neural ODE-evolved latent dynamics

CellFlows employs an encoder-decoder framework to extract a latent representation from input gene expression (Fig. 1). The encoder consists of three MLPs $Enc_z : \mathbb{R}^{2g} \to \mathbb{R}^l$, $Enc_t : \mathbb{R}^{2g} \to [0, 1]$, and $Enc_k : \mathbb{R}^{2g} \to \mathbb{R}^{3g}$. $Enc_z$, representing the inference network of the VAE, maps spliced and unspliced counts $x_i \in \mathbb{R}^{2g}$ for cell $i$ to its corresponding $l$-dimensional latent vector $z_i$ by sampling from the Gaussian distribution parameterized by $l$-dimensional mean $\mu_i(x_i)$ and standard deviation $\sigma_i(x_i)$. $Enc_t$ encodes a cell-specific developmental time $t_i \in [0, 1]$ corresponding to a continuous 'pseudotime' label that orders the cells along a lineage. $Enc_k$ encodes cell-gene-specific kinetic parameters corresponding to transcription rate $\alpha_i$, splicing rate $\beta_i$, and degradation rate $\gamma_i$, where $\alpha_i, \beta_i, \gamma_i \in \mathbb{R}^g$.

We apply an Euler method-based ODE solver to time-evolve the latent cell representations through pseudotime. We use an MLP $f_z : \mathbb{R}^l \to \mathbb{R}^l$ to represent the latent ODE function $\frac{dz}{dt} = f_z(z, t)$, where $t$ is pseudotime and $f_z$ outputs the transcriptomic vector field. Integrating the neural ODE is equivalent to solving an initial value problem (IVP) with initial condition $z_0$, where $z_0$ is set to the encoded latent representation of the cell assigned the earliest developmental pseudotime $t_0$. By solving the IVP through integration at all cell-labeled pseudotimes $\{t_i\}_{j=1}^n$, we generate ODE-predicted latent states for each cell $\{z_{t_i}\}_{i=1}^n$. Hence, we can directly compare each of our ODE-predicted cell latent states $z_{t_i}$ with a corresponding VAE-encoded latent representation of the measured cell $z_i$ with pseudotime $t_i$.

The decoder network $Dec_z \in \mathbb{R}^l \to \mathbb{R}^{2n}$, representing the generative component of the VAE, maps the $l$-dimensional latent representations to the original gene expression space $\mathbb{R}^{2n}$. The same $Dec_z$ is utilized both for reconstructing the original gene expression $x_i$ from $z_i$ and for decoding the ODE-predicted latent state $z_{t_i}$ to generate an ODE-predicted gene expression state $x_{t_i}$ for each cell $i$ and corresponding pseudotime $t_i$.

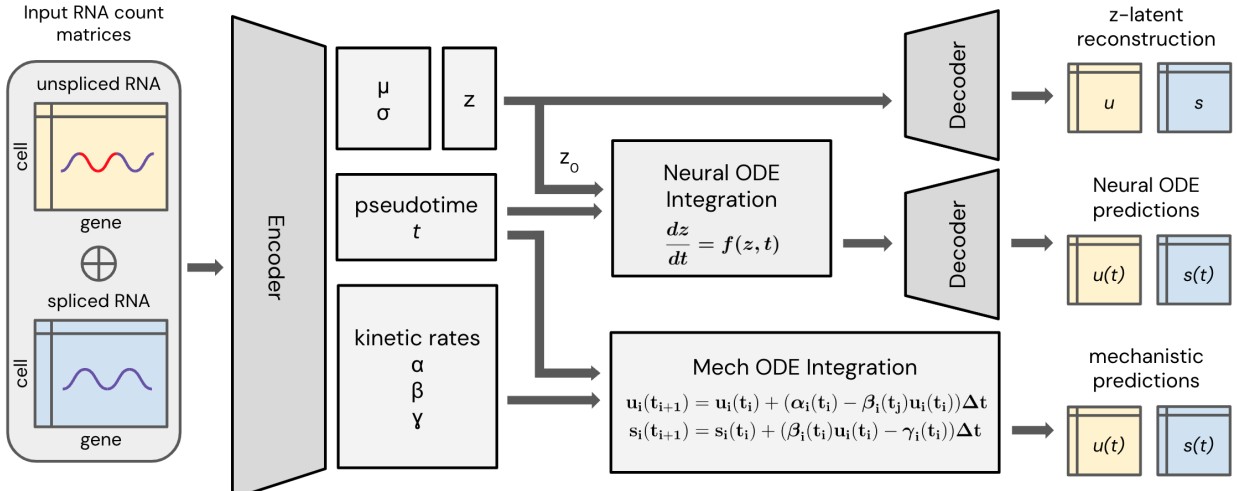

*Figure 1.* CellFlows jointly infers latent embeddings, developmental pseudotime, and kinetic rates. The VAE and neural ODE solver learn the latent state $z_i$ and its time-evolved counterpart $z_{t_i}$, where $t_i$ is the pseudotime label assigned to cell $i$. Both states are decoded to reconstruct gene expression matrices $x_i$ and $x_{t_i}$. $z_i$ is used to predict gene expression from the VAE-based latent states while $z_{t_i}$ is used to predict the observed gene expression at all future pseudotimes $t$ from an initial latent state $z_0$. Concurrently, the model uses kinetic equations to mechanistically time-evolve expression counts, leveraging the encoded splicing rates $\alpha_i$, $\beta_i$, and $\gamma_i$ to predict gene expression $x_{t_i}$ from an initial gene expression state $x_0$.

## 3.2. Interpretable mechanistic ODEs to infer cell-gene splicing dynamics

Concurrent with the VAE, we mechanistically predict gene expression using the linear ODEs described in Equation 1. The gradients of the spliced and unspliced abundances are parameterized by the kinetic rates $\alpha_i$, $\beta_i$, and $\gamma_i$ derived from $Enc_k$. By incorporating the Euler method-based ODE solver, we can predict the gene expression values across all cell-labeled pseudotimes $\{t_i\}_{i=1}^n$ via

$$u_i(t_{i+1}) = u_i(t_i) + (\alpha_i(t_i) - \beta_i(t_i)u_i(t_i))\Delta t$$
$$s_i(t_{i+1}) = s_i(t_i) + (\beta_i(t_i)u_i(t_i) - \gamma_i(t_i))\Delta t \quad (2)$$

where $\Delta t = t_{i+1} - t_i$ and $t_{i+1} > t_i$. We can solve the mechanistic linear ODEs at any given time point $t_i$ given an initial gene expression state $u_0$ and $s_0$, resulting in ODE-predicted spliced and unspliced abundances. The initial states $u_0$ and $s_0$ correspond to the spliced and unspliced abundances of the cell with the earliest pseudotime $t_0$. Simultaneously learning the dynamics from the linear ODEs in gene expression space and the neural ODE in latent space allows for mechanistic regularization of the neural dynamics. Additionally, the linear ODEs provide gene-cell-specific kinetic parameters that can identify genes undergoing significant transcriptional boosting and driving differentiation dynamics.

## 3.3. CellFlows Loss Function

The overall loss function is,

$$\mathcal{L}(\theta) = -\log p(x|z_{\text{VAE}}) + D_{KL}(q(z_{\text{VAE}}|x)||p(z_{\text{VAE}}))$$
$$+ ||z_{\text{VAE}} - z_{\text{ODE}}||_2^2 + ||x - \hat{x}_{\text{ODE}}||_2^2$$
$$+ ||x - \hat{x}_{\text{MECH}}||_2^2$$

where $\theta$ includes all trainable parameters and $x$ represents the input gene expression matrix with spliced and unspliced counts concatenated together. $z_{\text{VAE}}$ represents the VAE-encoded latent states of all measured cells, while $z_{\text{ODE}}$ is derived through neural ODE integration along learned developmental pseudotimes $t$ for each cell. $z_{\text{ODE}}$ is subsequently fed back into the decoder to produce ODE-predicted gene expression states $\hat{x}_{\text{ODE}}$. $\hat{x}_{\text{MECH}}$ is generated by utilizing the learned splicing kinetic parameters $\alpha, \beta, \gamma$ to parameterize and integrate the mechanistic ODEs as described in Equation 2.

The first two terms correspond to the standard VAE loss, with our data likelihood following a zero-inflated negative binomial. This distribution is parameterized by dispersion and dropout parameters output by the decoder $Dec_z$. The term $||z_{\text{VAE}} - z_{\text{ODE}}||_2^2$ represents the mean-squared error (MSE) between the VAE-encoded representations and the latent ODE-integrated predictions, serving to constrain excessive divergence between the two and encouraging the model to learn a vector field that best fits the cell trajectory. For the time-evolved gene expression matrices $\hat{x}_{\text{ODE}}$ and $\hat{x}_{\text{MECH}}$, we use the MSE to constrain the mechanistic

and decoded latent ODE-based predictions to align with the true gene expression $x$. Jointly learning the mechanistic and latent dynamics ensures that the inferred pseudotimes are consistent across different ODE frameworks, enabling accurate estimation of the cell-gene kinetics.

## 4. Results

We applied CellFlows and existing methods on the publicly available scRNA-seq datasets from (Maizels et al., 2023) to evaluate the performance of trajectory inference and RNA velocity estimation. The datasets consist of metabolically labeled data for mouse embryonic stem (ES) cells differentiating towards either a neural or mesodermal lineage. The sequencing data captures the development of mouse ES cells differentiating into neuromesodermal progenitors (NMPs), neural progenitors, mesodermal cells, and spinal cord neurons. Additionally, we evaluated all methods on an erythroid maturation dataset from (Pijuan-Sala et al., 2019), which captures the differentiation of hematoendothelial progenitor cells. All three benchmarking datasets include single-cell snapshots captured at multiple timepoints measured in days, allowing for an assessment of each model's pseudotime estimation.

For each dataset, we selected the top 2000 variable genes to ensure computational tractability. Unlike RNA velocity models such as veloVI and cellDancer, we explicitly chose not to perform $\log(1+x)$ transformation or $k$-nearest-neighbor smoothing to preserve the full biological signal from the measured gene expression. (Gorin et al., 2022) demonstrated that common preprocessing steps in single-cell analysis, such as normalization and smoothing, can introduce significant distortions and instabilities in RNA velocity estimates. By learning splicing dynamics in the VAE latent space, CellFlows demonstrates robustness through superior performance compared to methods that rely on extensive data preprocessing.

### 4.1. Trajectory Analysis and Visualization

In (Fig. 2), we provide an overview of the inferred pseudotime and vector field estimates for the neural branching dataset from (Maizels et al., 2023). The UMAP representations of the cells are derived from the weighted combination between the VAE-encoded cell states and the latent ODE-predicted states.

As is common practice in RNA velocity methods, we use our vector field to construct a cell-to-cell transition matrix based on nearest neighbors and visualize the trajectory with Uniform Manifold Approximation and Projection (UMAP) using the predicted transitions (Bergen et al., 2020). The UMAP primarily serves as a sanity check to confirm that the model has learned the appropriate differentiation trajectory.

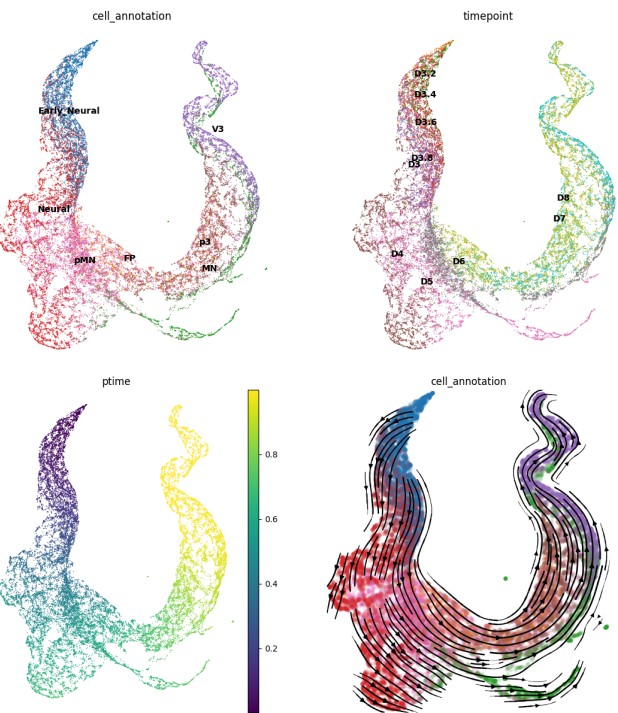

*Figure 2.* UMAP visualization of inferred developmental pseudotimes and vector field from gene expression of differentiating neural cells using CellFlows. The model projects the combined latent state from $z_{\text{VAE}}$ and $z_{\text{ODE}}$ and learns the vector field through the neural ODE.

Fig. 2 shows that the pseudotime estimates $t$ align with the true measured process times $\tau$. Early neural and neural cells are assigned earlier pseudotimes than V3 interneurons, motor neurons (MN), and floor plate (FP) cells. The inferred vector fields from CellFlows accurately predict the direction of differentiation and the transitions between cell types. The visualizations of weighted latent representations demonstrate a correct ordering of cell types along the differentiation pathway.

### 4.2. Quantitative Benchmarks

*Table 1.* Summary of methods based on whether they provide cell-gene-specific kinetic rates, accept unprocessed counts for robust inference, or perform joint inference of kinetics and pseudotime.

| METHOD | CELL-GENE | RAW COUNTS | JOINT INF. |
|---|---|---|---|
| CELLFLOWS* | $\checkmark$ | $\checkmark$ | $\checkmark$ |
| CELLDANCER | $\checkmark$ | $\times$ | $\times$ |
| VELOVI | $\times$ | $\times$ | $\checkmark$ |
| SCTOUR | $\times$ | $\checkmark$ | $\checkmark$ |

While all methods estimate a vector field over cells, previous works have primarily assessed the quality of their vector

*Table 2.* Spearman correlation coefficients between inferred pseudotime and marker gene expression across cell types.

| | NEURAL | | | | | | MESODERM | | | |
| METHOD | E-NEURAL | PMN | P3 | FP | MN | V3 | NMP | E-NEURAL | PMN | MESODERM |
| --- | --- | --- | --- | --- | --- | --- | --- | --- | --- | --- |
| CELLFLOWS* | **0.301** | 0.124 | 0.371 | 0.094 | **0.425** | **0.489** | 0.326 | 0.169 | 0.116 | 0.255 |
| CELLDANCER | 0.198 | **0.280** | 0.370 | 0.090 | 0.253 | 0.371 | 0.130 | 0.166 | 0.086 | 0.243 |
| VELOVI | 0.128 | 0.006 | 0.056 | 0.087 | 0.127 | 0.234 | 0.061 | 0.092 | **0.160** | 0.184 |
| SCTOUR | 0.230 | 0.164 | **0.398** | **0.132** | 0.416 | 0.248 | **0.360** | **0.175** | 0.044 | **0.258** |

fields based on UMAP visualizations. However, UMAPs are sensitive to hyperparameters such as the number of neighbors, and extensive tuning can introduce confirmation bias when interpreting results from RNA velocity models (Gorin et al., 2022). Given the unreliability of UMAP visualizations for robust and reproducible evaluation, we propose three quantitative benchmarks for assessing each method's performance on inferred estimates of pseudotime. In these benchmarks, we measure the Spearman correlation coefficient between the model-inferred pseudotimes and ground truth observations of process time points and cell type-specific marker gene expression.

We compare CellFlows to recently published methods: sc-Tour, cellDancer, and veloVI. veloVI was developed as a variational inference-based deep generative alternative to scVelo's dynamical expectation-maximization (EM) model (Gayoso et al., 2024). A summary of the methods based on their interpretable parameters and features is provided in Table 1. While all four methods provide pseudotime estimates, they vary greatly in their kinetics formulation. veloVI infers cell-gene-specific latent times jointly with gene-specific kinetic rates in their generative model. Due to veloVI's biophysical assumption of uniform gene kinetics across cells, however, the method cannot be used to determine the main drivers of cell subpopulations. cellDancer relies on a two-stage training process where the latent time is inferred separately from the deep learning model that learns RNA velocity. CellFlows jointly infers pseudotime and cell-gene-specific kinetics through its combined neural and mechanistic ODE framework.

*Table 3.* Spearman correlation coefficients between inferred pseudotime and process time $\tau$.

| METHOD | ERYTHROID | NEURAL | MESODERM |
| --- | --- | --- | --- |
| CELLFLOWS* | **0.816** | 0.802 | **0.920** |
| CELLDANCER | 0.739 | 0.738 | 0.765 |
| VELOVI | 0.605 | 0.279 | 0.662 |
| SCTOUR | 0.791 | **0.846** | 0.917 |

**Time Correlations.** We can benchmark the quality of inferred pseudotimes across methods by examining how well aligned their pseudotimes are with the process time by cal-

culating the Spearman correlation coefficient between the two quantities. As shown in Table 3, CellFlows outperforms the other methods in pseudotime correlations in two of three datasets, and is second only to scTour for the neural dataset.

**Marker Gene Expression Correlation.** We can further benchmark the quality of inferred pseudotimes from each method by examining how well correlated their pseudotimes are with the expression of marker genes per cell type in each dataset. Cell-type specific markers are only available for the neural and mesoderm datasets from (Maizels et al., 2023). We calculated the Spearman correlation coefficient between the marker gene's expression profile and the inferred pseudotime for each annotated cell type population. We then averaged the coefficients across the marker genes to represent the method's overall performance for each cell type.

CellFlows and scTour closely match in performance in marker gene expression correlations across cell types while both methods mostly outperform cellDancer and veloVI (Table 2). CellFlows shows the most improvements for Early Neural and V3 cells in the Neural dataset over the competing methods, and scTour performs best overall in the Mesoderm dataset.

## 5. Discussion

In this work, we introduced CellFlows, a novel computational framework that combines neural ordinary differential equations with mechanistic modeling to jointly infer cell-gene-specific splicing kinetics, pseudotimes, and vector fields from single-cell transcriptomic data. Through UMAP visualizations and quantitative benchmarking, we demonstrated CellFlows' ability to accurately estimate pseudotime values that align closely with measured time points and marker gene expression patterns across cell subpopulations. A key strength of our approach is the integration of data-driven neural networks with interpretable mechanistic equations describing the underlying biological processes.

The implications of CellFlows extend beyond RNA velocity and single-cell genomics analyses. Our framework provides a powerful and flexible paradigm for studying latent dynamical systems. Any domain where complex dynamics can be partially described by mechanistic equations, while also ben-

efiting from flexible neural network representations, could leverage the strengths of our combined modeling approach. Future research could explore extensions of CellFlows to handle more complex and higher-dimensional dynamics, incorporate additional data modalities beyond transcriptomics, or enable transfer learning across related latent dynamical systems. Ultimately, CellFlows represents a promising step towards tighter integration of neural networks with differential equations and domain knowledge for powerful scientific discovery.

## Acknowledgements

This material is based upon work supported by the National Science Foundation Graduate Research Fellowship under Grant No. 2036197. This work was also made possible by support from the MacMillan Family and the MacMillan Center for the Study of the Non-Coding Cancer Genome at the New York Genome Center.

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
