# OpenReview forum: "CellFlows: Inferring Splicing Kinetics from Latent and Mechanistic Cellular Dynamics"
_ICML.cc/2024/Workshop/ML4LMS — ML4LMS Poster_

### Official Review · Reviewer_ovX9 · 2024-06-12
**A valuable contribution to the field**

**Rating:** 7
**Confidence:** 4

**Review:**

## Summary

The proposed method, CellFlows, is a novel architecture designed to infer gene and context-specific kinetic rates at single-cell resolution by incorporating self-supervised neural dimensionality reduction and neural-based latent time estimation into a mechanistic model. This improves model interpretability and accuracy in modeling splicing dynamics, helping to identify linear and branching cellular differentiation pathways in mouse embryonic stem cells.

## Evaluation

**Quality:** The paper is of high quality, presenting a comprehensive and well-validated approach to improving the inference of cellular dynamics from single-cell RNA sequencing data. The methodology is robust, and empirical results support the claims made.

**Clarity:** The clarity of the paper is good, with detailed explanations of the proposed methods and experimental setups

**Originality:** The integration of self-supervised neural dimensionality reduction and neural ODE-evolved latent dynamics into a mechanistic model is a novel approach that addresses key challenges in inferring splicing kinetics and cellular dynamics.

**Significance:** The work is significant for improving the understanding of cellular dynamics and gene expression regulation. The method's applicability to various datasets and its superior performance compared to existing methods highlight its potential impact on the field of single-cell genomics.

## Pros and Cons:

**Pros:**

- The proposed method is novel, which combines neural networks with mechanistic models to improve inference of cellular dynamics.
- The method Demonstrates superior performance in inferring pseudotimes and gene-specific kinetics compared to existing methods.
- The method is applicable to various datasets, providing a powerful tool for studying latent dynamical systems.

**Cons:**

- The combination of VAE and neural ODEs with mechanistic modeling may require significant computational resources and expertise.
- The performance of the model depends on the quality of the single-cell RNA sequencing data.

---

### Official Review · Reviewer_q1dM · 2024-06-12
**Reviewing CellFlows**

**Rating:** 7
**Confidence:** 4

**Review:**

Authors present a new workflow to track cellular developmental trajectories.

---

### Official Review · Reviewer_8by8 · 2024-06-12

**Rating:** 8
**Confidence:** 4

**Review:**

Summary:
The authors introduce a VAE combined with a neural ODE to infer cell state dynamics from spliced and unspliced scRNA expression values. The introduced architecture is novel for cell dynamic prediction and represents a jointly modeled latent state capable of predicting cellular expression states across time from a mechanistic, latent, and neural ODE perspectives.

Strengths and weaknesses:
The method outperforms benchmarks across most cell types found in two datasets. The flow and problem outline of the paper is clear. The results are presented clearly and Figure 1 outlines the CellFlows' architecture.
From these results, the need for joint inference across the CellFlows' architecture is not convincing. The study would benefit from an ablation study of the model's architecture. Additionally, the paper would benefit from additional downstream analysis of predictions on relevant datasets to highlight practical applicability.

Overall, I think the paper aligns with the workshop's central theme and has clear applications in the life sciences through a machine learning driven approach.

---

### Official Review · Reviewer_PJX3 · 2024-06-12
**Interesting work, more fine-grained analysis needed**

**Rating:** 6
**Confidence:** 3

**Review:**

This work seeks to unite neural network-based and mechanistic (differential equation) models to jointly infer RNA splicing kinetics. To this end, the proposed model consists of 3 MLP encoders that take RNA read counts and encode them into a latent vector, a scalar pseudotime and kinetic parameters, the latter two of which are then used to solve a neural ODE (for predicting future states of the latent vector) and a mechanistic ODE (w.r.t. gene expression kinetics). Results show that the proposed model is competitive in terms of spearman correlation between inferred pseudotime and the expression profile of marker genes as well as process time.

This is a work-in-progress that seems promising. For the full paper, at least the following will be needed:
- Experiments on more datasets and analysis on the relationship to functional properties;
- Uncertainty analysis;
- Possibly a more sophisticated mechanistic mode.